

# Adipose tissues of MPC1± mice display altered lipid metabolism-related enzyme expression levels

Shiying Zou[1,2], Liye Zhu[1,2], Kunlun Huang[1,2], Haoshu Luo[3], Wentao Xu[1,2] and Xiaoyun He[1,2]

[1] China Agricultural University, Beijing Advanced Innovation Center for Food Nutrition and Human Health, College of Food Science and Nutritional Engineering, Beijing, China
[2] Key Laboratory of Safety Assessment of Genetically Modified Organism (Food Safety), Ministry of Agriculture, Beijing, China
[3] China Agricultural University, College of Biological Sciences, Beijing, China

## ABSTRACT

Mitochondrial pyruvate carrier 1 (MPC1) is a component of the MPC1/MPC2 heterodimer that facilitates the transport of pyruvate into mitochondria. Pyruvate plays a central role in carbohydrate, fatty, and amino acid catabolism. The present study examined epididymal white adipose tissue (eWAT) and intrascapular brown adipose tissue (iBAT) from MPC1± mice following 24 weeks of feeding, which indicated low energy accumulation as evidenced by low body and eWAT weight and adipocyte volume. To characterize molecular changes in energy metabolism, we analyzed the transcriptomes of the adipose tissues using RNA-Sequencing (RNA-Seq). The results showed that the fatty acid oxidation pathway was activated and several genes involved in this pathway were upregulated. Furthermore, qPCR and western blotting indicated that numerous genes and proteins that participate in lipolysis were also upregulated. Based on these findings, we propose that the energy deficiency caused by reduced MPC1 activity can be alleviated by activating the lipolytic pathway.

## INTRODUCTION

Pyruvate is a central substrate in energy metabolism, paramount to carbohydrate, fatty acid, and amino acid catabolic and anabolic pathways (*Compan et al., 2015*). MPC1 is a component of the MPC1/MPC2 complex that facilitates the transport of pyruvate into mitochondria. Previous studies have shown that complete MPC1 deficiency results in embryonic lethality in mice (*Li et al., 2016*; *Vanderperre et al., 2016*). *Rauckhorst et al. (2017)* found that a high-fat diet increased hepatic mitochondrial pyruvate utilization and tricarboxylic acid (TCA) cycle capacity in a hepatocyte MPC disruption model. *Vanderperre et al. (2016)* generated a mouse strain with complete loss of MPC1 expression, which exhibited such severe energy deficit that no mouse could survive. However, these phenotypic consequences were reversed when the pregnant dams were fed a ketogenic diet. Fatty acid metabolism plays a particularly important role in energy balance when MPC is

Corresponding author
Xiaoyun He, raininghe@163.com

lost or inhibited, yet little research has examined the molecular changes involved. In our previous study, a mouse line with an MPC1 protein partial deficiency was generated using the CRISPR/Cas9 system. MPC1$^{\pm}$ mice experienced energy deficiency, as evidenced by low body weight, decreased movement, low body shell temperature, and limited adipose accumulation (*Zou et al., 2018*). We hypothesized that fatty acid oxidation alleviates the energy deficiency; however, the molecular changes in energy metabolism remained unclear. Therefore, the present study explored the regulatory mechanism underlying fatty acid metabolism in MPC1$^{\pm}$ mice.

Adipose tissues play a central role in regulation of energy balance. Adipose tissues are not only organs that store energy in the form of fatty acids but synthesize and secrete numerous growth factors, enzymes, and hormones that are related to energy homeostasis (*De Jong et al., 2015*; *Yang et al., 2017*). Various metabolic processes are regulated by adipose tissues, including glucose homeostasis, lipid metabolism, and inflammation (*Xu et al., 2018*). The two main types of adipose tissues in mammals are white adipose tissues (WATs) and brown adipose tissues (BATs). The white adipocytes are round-shaped cells that contain a single large fat droplet, whereas brown adipocytes are generally smaller in size (*Lo & Sun, 2013*; *Spiegelman & Flier, 2001*). Epididymal white adipose tissues (eWATs) that are attached to the epididymis and testis are the largest visceral WATs and are referred to as a 'classical' white fat depots (*De Jong et al., 2015*).

RNA-Sequencing (RNA-Seq) is a well-developed approach to study transcriptome profiling that uses deep-sequencing technologies. It provides a much more precise measurement of transcripts and isoforms than other methods (*Wang, Gerstein & Snyder, 2009*). In this study, RNA-Seq was utilized to detect changes in the energy pathway of MPC1$^{\pm}$ mice. We subsequently characterized changes in the fatty acid metabolism of MPC1$^{\pm}$ mice by detecting alterations in various genes and proteins that are involved in the energy metabolism pathway to better understand the role of MPC1.

## MATERIAL AND METHODS

### MPC1$^{\pm}$ mice and wild-type (WT) mice were fed for 24 weeks

Six male MPC1$^{\pm}$ mice and six male WT C57BL/6 mice were fed for 24 weeks with sterile water and food *ad libitum*. These were used in the experiments after weaning (at 5 weeks), with six mice per group. Three mice were housed in one cage. This research was conducted at the SPF Animal Laboratory of the Supervision & Testing Center for GMO Food Safety, Ministry of Agriculture (Beijing, China), with the license number SYXK (Beijing) 2015-0045. The environment temperature was maintained at 20 °C–24 °C, with humidity between 40%–70%, and a 12-h light/dark cycle. During the acclimatization period, the animals were fed with commercially produced standard laboratory animal chow manufactured by Keao Xieli Li Feed Co., Ltd. (Beijing, China).

All experimental procedures were performed according to guidelines provided by the Animal Welfare Act and Animal Welfare Ordinance. The animal experiment and housing procedures were conducted in compliance with the OECD Good Laboratory Practice guidelines. This animal study was approved by the Animal Experimental Welfare & Ethical

Inspection Committee (No. 2016005), the Supervision & Testing Center for GMO Food Safety, Ministry of Agriculture (Beijing, China). The animals were handled according to the Guide for the Care and Use of Laboratory Animals (*Bayne, 1996*), and the attendant committees approved all protocols utilized.

## Tissue collection

Orbital sinus blood was sampled from 10-week-old mice at after 5 h or 48 h of fasting, and centrifuged at 4,000 g for 10 min. The isolated sera were stored at −80 °C until further analysis. Serum triglycerides (TG) and non-esterified fatty acids (FFAs) were measured using an ELISA kit (Beijing Fangchengjiahong Technology Co., Ltd, Beijing, China).

After 24 weeks of feeding, the mice were sacrificed. The iBATs and eWATs were collected and weighed. Interscapular BATs are visible at the level of the shoulder blades when the back skin is removed. Tissues were fixed in 4% paraformaldehyde at room temperature overnight. The fixed tissues were subsequently dehydrated by a graded series (70% to 95%) of ethanol and then embedded in paraffin. Paraffin sections of 5 μm thickness were stained with hematoxylin and eosin (H&E). Portions of the iBATs and eWATs were frozen in liquid nitrogen until further analysis.

## RNA isolation, cDNA synthesis, and real-time qPCR

Total RNA was extracted from the eWATs and iBATs using TRIzol according to the manufacturer's instructions. RNA was reverse transcribed using a High-Capacity cDNA Reverse Transcription Kit (AH341; TransGen, Illkirch-Graffenstaden, France), following the manufacturer's protocol. The qPCR reactions were conducted with TransStart Green qPCR Super mix kit (AQ101; Transgen, Illkirch-Graffenstaden, France) on a CFX96 system (Bio-Rad, Richmond, CA, USA). RNA concentrations of each sample were determined using a NanoDrop 1000 system (Thermo Scientific, Waltham, MA, USA). All RNA samples were within a 260:280 ratio >1.8, and a 260:230>1.8. Relative target mRNA abundance was normalized to that of *GADPH*. Primer sets for quantitative real-time PCR are summarized in Table 1.

## RNA-Seq
### Total RNA extraction
Total RNA was extracted using RNeasy Micro Kit (74004; QIAGEN, Valencia, CA, USA), according to the manufacturer's instructions. Extracted RNA was quantified with the Qubit RNA Assay Kit (Invitrogen).

### Library generation and sequence
mRNA library construction was performed with an NEBNext Ultra RNA Library Prep Kit for Illumina (E7530; New England Biolabs, Ipswich, MA, USA), according to the manuals, using 500 ng total RNA. Briefly, library preparation was performed using the following steps: RNA fragmentation, reverse transcription using random primers, second strand cDNA synthesis, end repair, dA-tailing, adapter ligation, U excision, and PCR enrichment. The libraries were sequenced on Illumina Hiseq X Ten instruments with 150-bp paired-end reads. All clusters that passed the quality filter were exported as FASTQ files.

 

**Table 1** Sequences of primers used for qPCR analysis.

| Gene name | Forward(5′–3′) | Reverse (5′–3′) |
| --- | --- | --- |
| CPT2 | CAGCACAGCATCGTACCCA | TCCCAATGCCGTTCTCAAAAT |
| ATGL | CCAACACCAGCATCCAGT | CAGCGGCAGAGTATAGGG |
| HSL | CGCCATAGACCCAGAGTT | TCCCGTAGGTCATAGGAGAT |
| Cox4 | CGGCGTGACTACCCCTTG | TGAGGGATGGGGCCATACA |
| Perilipin2 | GATTGAATTCGCCAGGAAGA | TGGCATGTAGTCTGGAGCTG |
| Perilipin3 | CTGAGAAAGGCGTCAAGACC | TTTCTTGAGCCCCAGACACT |
| PGC1$\beta$ | CGTATTTGAGGACAGCAGCA | TACTGGGTGGGCTCTGGTAG |
| MPC1 | GACTATGTCCGGAGCAAGGA | TAGCAACAGAGGGCGAAAGT |
| MPC2 | TGTTGCTGCCAAAGAAATTG | AGTGGACTGAGCTGTGCTGA |
| FASN | TCCAAGACTGACTCGGCTACTGAC | GCAGCCAGGTTCGGAATGCTATC |
| ACC | AGCTGATCCTGCGAACCT | GCCAAGCGGATGTAAACT |
| PPAR$\alpha$ | ATACATAAAGTCCTTCCCGCTG | GGGTGATGTGTTTGAACTTGATT |
| PPAR$\beta$ | GCTATCATTACGGAGTCCACG | TCGCACTTGTCATACACCAG |

*Bioinformatics analyses*

After trimming the adaptor sequences and removing low-quality reads from raw RNA-Seq using Cutadapt (v1.10), the reads were aligned to the mm10 reference genome using Tophat2 (v2.0.13), and the reads aligned to genes were counted using Cufflinks (v2.2.1). The FPKMs were normalized using Cuffnorm. Differentially expressed genes (DEGSs) were calculated using Cuffdiff. This work was accomplished by Beijing Geek Gene Technology Co., Ltd (Beijing, China).

Unsupervised hierarchical clustering was conducted using $\log_2(\text{FPKM}+1)$ across all samples. Genes used for clustering were selected according to maximum$\{\log_2(\text{FPKM}+1)\}>1$ and sd$\{\log_2(\text{FPKM}+1)\}>0.5$. Each group comprised three individuals (greater than twofold difference in xpression; $P<0.05$). An MA-plot, an application of a Bland-Altman plot for visual representation of genomic data, was used to visualize the differences between the measurements taken in two samples by transforming the data into M (log ratio) and A (mean average) scales, and then plotting these values. Each point on the plot stands for a gene. The $x$-axis was calculated as $\log_2((\text{FPKMA}+1)/(\text{FPKMB}+1))$, whereas the $y$-axis was calculated as $1/2\ [\log_2(\text{FPKMA}+1)+\log_2(\text{FPKMB}+1)]$. DEGs were marked in red ($P \leq 0.001$ and FC $\geq 1.2$) and in blue ($0.05 \geq P>0.001$ and FC $\geq 1.2$). A Volcano plot, which is a scatter-plot that is used to readily identify changes in large data sets composed of replicate data, was used to indicate the significance vs. fold-change on the $y$ and $x$ axes, respectively. Each dot stands for a gene. The $x$-axis was calculated as $\log_2((\text{FPKMA}+1)/(\text{FPKMB}+1))$, and the $y$-axis was calculated as log10(p). Pathways and molecule functions that were enriched in the MPC1$^{\pm}$ mice were identified using GO term and KEGG pathway analyses.

## Western blotting

The eWATs and iBATs were used for protein isolation and quantification. The tissues proteins were extracted using a RIPA lysis buffer (0.15 M NaCl, 1.0% Triton X-100, 0.5% sodium deoxycholate, 0.1% SDS, 50 mM Tris (pH 7.4) 0.1 M EDTA, 2 mg/L leupeptin,

and 100 mg/L sodium fluoride). Proteins were isolated by centrifugation (4 °C, 12,000 g, 15 min), and protein concentrations were measured with a BCA assay kit (Beyotime Biotechnology, China). Equal amounts of protein were separated by 12% SDS-PAGE for ATGL, 8% SDS-PAGE for ACC, 10% Tricine-PAGE for MPC1 and MPC2 (*Haider, Reid & Sharp, 2012*), and the proteins were subsequently transferred to PVDF using the Mini-PROTEAN® Tetra Vertical Electrophoresis Cell (Bio-Rad Laboratories, Hercules, CA, USA) (80 V, 60∼120 min for SDS-PAGE, and 150V, 60∼100 min for Tricine-PAGE). The filters were incubated for 1.5 h in Blotto solution (5% milk powder (w/v) in TBST and 3.2 mM $MgCl_2$, pH 7.4). The membranes were incubated overnight with primary antibodies at 4 °C and then probed with secondary antibodies conjugated with horseradish peroxidase HRP. Primary antibodies used in this study were rabbit mAb ACC (Cell Signaling Technology 3662), rabbit mAb ATGL (2138; Cell Signaling Technology, Danvers, MA, USA), rabbit mAb $\beta$-Tubulin (2146; Cell Signaling Technology, Danvers, MA, USA), rabbit MPC1 (14462; Danvers, MA, USA), and rabbit MPC2 (46141; Cell Signaling Technology, Danvers, MA, USA) at a 1:1,000 dilution. Secondary antibodies were anti-rabbit antibodies (#A0208; Beyotime Biotechnology, Jiangsu, China) at a dilution of 1:2,000. Western blot detection was performed using chemiluminescent HPR substrate (Millipore Corporation, Billerica, MA, USA) and subsequently, autoradiography was performed with a ChemiScope 3300 mini Imaging and Analysis System (Clinx Science Instruments Co., Ltd., Shanghai, China). Quantification of the immunoblot was performed by grayscale value analysis with the Clinx image analysis software. The relative protein expression data were normalized to that of $\beta$-tubulin.

## Data analysis

Microsoft Excel was used to organize and statistically analyze the data and prepare figures. Microsoft PowerPoint was used to draw the roadmap on energy metabolism. Unless otherwise noted, data were represented as the mean ± SD, and statistical significance was determined using a two-tailed student's *t-test* with a statistically significant difference defined as a *P* value <0.05. Differential gene expression and gene-set enrichment analysis were analyzed using the KEGG pathway.

## RESULTS

### MPC1 deficiency increases fat depletion and changes the pathomorphism of eWAT

To investigate how metabolism is reprogrammed in response to MPC1 deficiency in vertebrates, we generated a mouse model (MPC1$^\pm$) using the CRISPR/Cas9 system (*Zou et al., 2018*). Figure 1A shows that after 24 weeks of feeding, the body weight and eWAT weight of the MPC1$^\pm$ mice significantly decreased compared to the WT mice (*P<0.05*). The weight of the iBATs of the MPC1$^\pm$ mice also decreased but was not statistically significant. We isolated the eWATs and iBATs of the MPC1$^\pm$ mice to determine relative *Mpc* mRNA and MPC protein expression. The qPCR results (Figs. 1B and 1C) showed that the expression of *mpc1* significantly decreased in the iBATs and eWATs. Western blotting of the eWATs and iBATs in the heterozygous (MPC1$^\pm$) mice demonstrated a significant reduction (*P<0.01*)

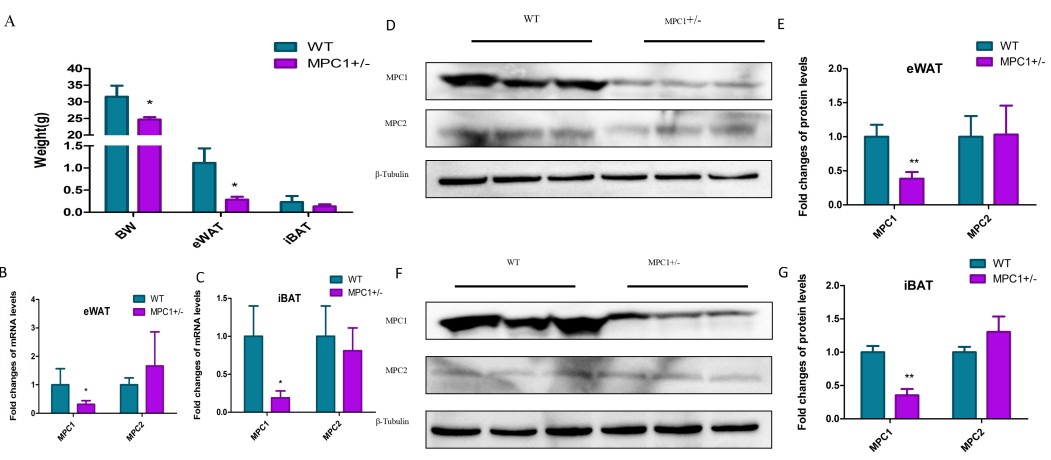

**Figure 1  MPC mRNA and protein expression levels in MPC1± mice.** (A) The body weight (BW), eWAT weight and iBAT weight in the 24-week-old MPC1± mice and WT mice, $n = 6$; (B) The fold-changes of mRNA levels in eWATs; (C) The fold changes of mRNA levels in iBAT; (D) MPC1 and MPC2 protein expression levels in eWATs; (E) The relative expression level of MPC1 and MPC2 protein in eWATs; (F) MPC1 and MPC2 protein expression levels in iBATs; (G) The relative expression level of MPC1 and MPC2 protein in iBATs; Data are expressed as the mean ± standard deviation (mean ± SD). The relative protein expression data were normalized to that of $\beta$-tubulin. *$P < 0.05$, **$P < 0.01$ for MPC1± mice *vs*. WT mice.

in MPC1 protein expression compared to the WT (Figs. 1D–1G). Despite MPC1 deficiency, normal levels of *mpc2* mRNA and MPC2 protein expression were observed.

Histopathological analysis indicated that the volume of adipocytes in the MPC1± mice was smaller than that in WT mice (Fig. 2A). Small and tight adipocytes were observed in the eWATs of the MPC1± mice. We measured the diameter of 100 adipocytes (Fig. 2B) and calculated the diameter percentage (Fig. 2C). The adipocyte diameter of the MPC1± mice were within the range of 20~30 μm, whereas that of the WT mice was 40~50 μm. The eWAT mass, which was calculated according to the size and the number of adipocytes, was significantly correlated with eWAT weight and average adipocyte size. Adipocytes play a central role in energy balance, in which these serve as major sites of storage and expenditure, and as endocrine cells, secrete adipokines and other molecules that regulate energy storage and metabolism of other tissues (*Spiegelman & Flier, 2001*). Both body fat accumulation and adipocyte size are associated with metabolic abnormalities. Adipocyte size is thought to be related to its physiological function and has been positively correlated with TG levels.

Figures 2D and 2E show that after 5 h or 48 h of fasting, serum FFA and TG levels increased in the MPC1± mice compared to the WT mice. The MPC1± mice displayed energy accumulation deficiency. Fatty acids are stored in the form of TG in all cells, but predominantly within adipose tissue such as the iBATs and eWATs. When the body acquires excessive amounts of energy from food, the surplus energy is stored in adipose tissues. In response to energy demands, the fatty acids from stored TGs are mobilized for use by peripheral tissues. In the MPC1± mice, the eWATs displayed reduced accumulation

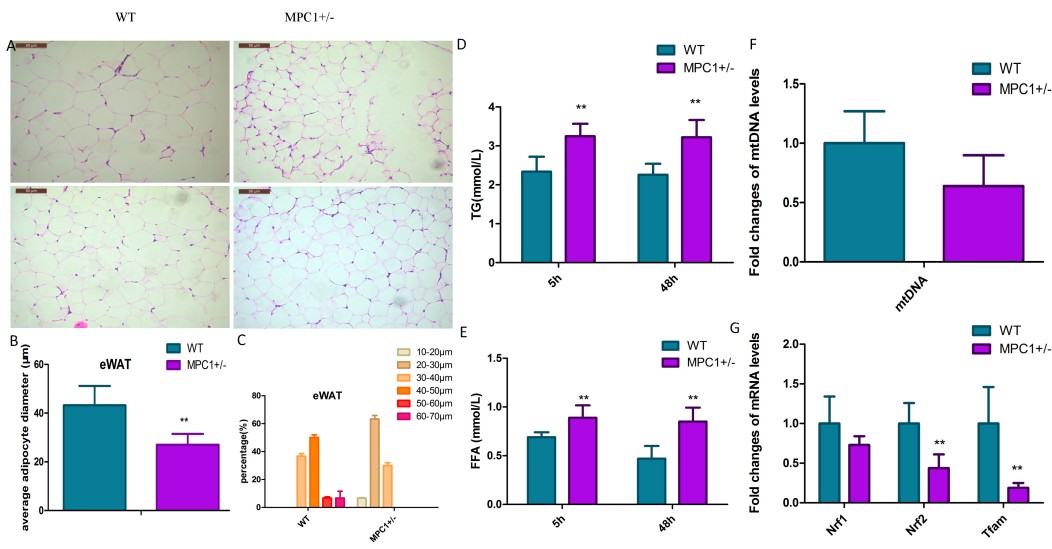

**Figure 2 Histopathological changes and mitochondrial markers of eWATs in the MPC1$^\pm$ mice.** (A) Representative HE-stained images of eWATs from 24-week-old WT and MPC1$^\pm$ mice, scale bars: 50 $\mu$m, $n = 6$; (B) Average adipocyte volume of eWATs in WT and MPC1$^\pm$ mice, $n = 6$; (C) The percentage of adipocyte volume in picture A ($n = 100$ adipocyte). The volume (diameters) was tested by the software installed in the pathological microscope (RM2500, Leica), 100 adipocytes in each mouse were tested. (D) Serum TG levels of TG in mice that underwent 5 h and 48 h fasting, $n = 6$; (E) Serum FFA levels of mice that underwent 5 h and 48 h fasting, $n = 6$; (F) Fold-changes in mtDNA levels in eWATs, $n = 6$; (G) The mRNA expression levels of *Nrf1*, *Nrf2*, and *Tfam* in the eWATs, $n = 6$. * $P < 0.05$, ** $P < 0.01$ for MPC1$^\pm$ mice *vs.* WT mice.

of lipid droplets. These findings suggest that the TGs are metabolized for use by peripheral tissues.

The mitochondrial damage was evaluated through the mtDNA number and the mRNA expression levels of *nuclear respiratory factors 1/2* (*Nrf1*, *Nrf2*) and *mitochondrial transcription factor A* (*Tfam*). The mtDNA number in the eWATs decreased but was not statistically significant (Fig. 2F). The mitochondrial biogenesis genes (*Nrf2* and *Tfam*) were downregulated in the MPC1$^\pm$ mice relative to those of the WT mice, suggesting that MPC1$^\pm$ mice have reduced mitochondrial biogenesis activity (Fig. 2G).

## The fatty acid metabolism pathway is activated in MPC1$^\pm$ mice

The primary sources of fatty acids for oxidation include the diet or those mobilized from adipose tissues. Therefore, changes in adipose tissues were further investigated in this study. Because the eWATs displayed significant changes in the weight and histopathological analysis, we subsequently conducted RNA-Seq analysis. A *p*-value threshold of <0.05 and a 1.2-fold change were adopted to define significant biological variations. The eWATs of the MPC1$^\pm$ mice showed 1,625 differentially expressed genes relative to the WT mice, which included 880 upregulated genes and 745 downregulated genes. Unsupervised hierarchical clustering was conducted using log$_2$(FPKM+1) across samples (Fig. 3A). All 1,625 genes were used in clustering, which were selected using a maximum {log$_2$(FPKM+1)}>1 and sd{log$_2$(FPKM+1)}>0.5. Expression values are represented in different colors, indicating

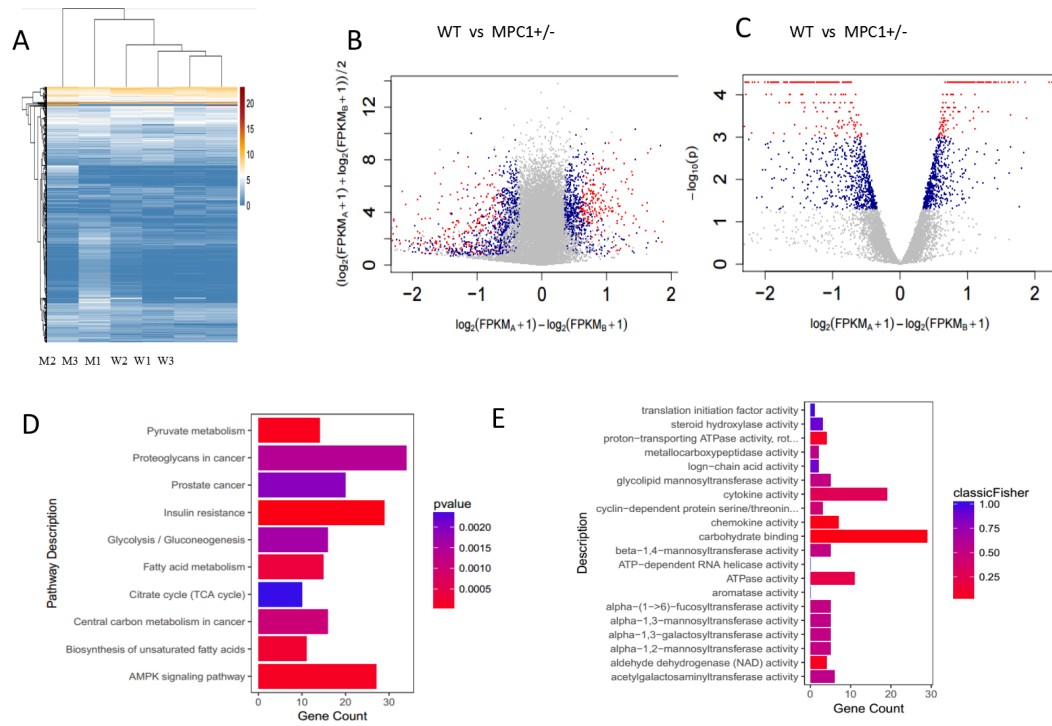

**Figure 3** **Transcriptome analysis of gene expression profile mRNAs in eWATs.** (A) Unsupervised hierarchical clustering; (B) MA-plot of genes sequenced in eWATs of the MPC1$^{\pm}$ mice and WT mice. Differentially expressed genes are marked in red (*P* $\leq$*0.001 and FC* $\geq$*1.2*) and blue (*0.05* $\geq$*P>0.001 and FC* $\geq$*1.2*); (C) Volcano plot of genes sequenced in eWAT. Groups and colors are set as that of MA-plot. (D, E) Pathways (D) and molecular function (E) enriched in MPC1$^{\pm}$ mouse eWATs as compared to the WT *n* = 3.

expression levels above or below the median expression level across all samples. The MA-plot and the volcano plot show the fold changes of significantly differentially expressed genes (Figs. 3B and 3C). Significance versus fold-change were plotted on the *y* and *x* axes, respectively. Each dot stands for a gene. We further determined the genes and pathways that were differentially expressed between the MPC1$^{\pm}$ mice and WT mice. Pathways and molecular functions that were enriched in MPC1$^{\pm}$ mice compared to the WT mice were identified using GO and KEGG analyses (Figs. 3D and 3E). Various pathways were activated in the MPC1$^{\pm}$ mice, including carbon metabolism, AMPK signaling pathway, insulin signaling pathway, oxidative phosphorylation, pyruvate metabolism, fatty acid metabolism, TCA cycle, and cancer pathway. Some of the genes corresponding to fatty acid metabolism are presented in Table 2 and Table S1.

As expected, the pathway of fatty acid oxidation in acyl-CoA degradation was activated. Several genes involved in fatty acid oxidation were upregulated, including *acetyl-coenzyme A acetyltransferase 2* (*Acat2*), *enoyl-coenzyme A, hydratase coenzyme A dehydrogenase* (*Ehhadh*), *peroxisome proliferator-activated receptor* α (*Pparα*), *fatty acid desaturase 1* (*Fads1*), *acyl-CoA synthetase* (*Acs*), *alcohol dehydrogenase 1* (*Adh1*), *fatty acid transport proteins* (*Fatps*), *enoyl coenzyme A hydratase,* and *short chain 1* (*Echs1*). ACAT2 is the major
**Table 2   Signaling pathway enriched in eWAT of MPC1$^{\pm}$ mice in RNA-Seq.**

| Signaling pathway | Representative genes | Clustering for analysis |
|---|---|---|
| Adipocytokine signaling pathway | Acac$\beta$, Irs1, Ppar$\alpha$, Ppar$\gamma$, Rxr$\beta$ Acsbg1 | |
| Fatty degradation | Acat2, Adh1, Ehhadh, Echs1, Fads1, FATPs | |
| Cholesterol metabolism | Lrp2, Ldlr, Lipg, Lrpap1 | Lipolysis |
| Regulation of lipolysis in adipocytes | Adora1, Irs1, Pik3r1, Pik3c$\beta$ | |
| Oxidative phosphorylation | Atp5$\alpha$1, Atp5c1, Cox4, Cox7a, Ndufv1, Ndufs2, Nduf$\beta$9, Nduf$\alpha$/ $\beta$1, Sdhd, Sdha | |
| Fatty acid digestion and absorption | Acat2, Dgat1, Scarb1, Agpat2 | Lipogenesis |
| Steroid biosynthesis | Cyp51, Lss, Nsdh1, Cyp2e1, Hsd11$\beta$1, Hsd17$\beta$12 | |
| Insulin signaling pathway | Gys1, Irs1, Pik3r1, Ppar$\alpha$, Ppp1r3$\beta$, Ppp1r3c, Pik3c$\beta$ | Insulin resistance |
| Pathway in cancer | Axin2, Esr1, Gstm5, Rxrb, Tcf7, Wnt4, Wnt2b, Glut1, Tgf$\alpha$, Ccnd1, Cdk4/6, Rxr$\beta$ | Cancer |

cholesterol-esterifying enzyme that plays a critical role in preventing murine atherosclerosis and hypertriglyceridemia (*Alger et al., 2010*). *Ehhadh* is part of the classical peroxisomal fatty acid $\beta$-oxidation pathway, which is highly inducible via peroxisome proliferator-activated receptor $\alpha$ (PPAR$\alpha$) activation (*Bjorndal et al., 2018*). Strong associations between *fads1* and blood fatty acid levels have been reported, particularly that between *fads1* and PUFAs and long-chain PUFAs (*Glaser, Heinrich & Koletzko, 2010*). The expression level of *Acs*, a gene involved in fatty acid metabolism, was found to be associated with increased lipid loading as well as higher insulin sensitivity. *Adh1* plays an important role in ethanol oxidation and fatty acid degradation (*Lieber, 2004*). *Echs1*, a gene involved in fatty acid beta oxidation, regulates cellular ATP production (*Zhu, Xi & Kukreja, 2012*). The *fatps* gene facilitates the uptake of very long-chain (VLCFA) and long-chain fatty acids (LCFA) (*Guitart et al., 2014*).

Oxidative phosphorylation is the metabolic pathway in which cells use enzymes to oxidize nutrients, thereby releasing energy, which is used to produce ATP. In transcriptomics, oxidative phosphorylation was found to be activated in the MPC1$^{\pm}$ mice. The *PPAR* signaling pathway was activated, which was significant because *PPAR$\alpha$* and *PPAR$\gamma$* can promote fatty acid oxidation (*Biswas et al., 2016*). The oxidative phosphorylation of mitochondria was upregulated, including genes in the ATP synthase (*Atp5a1, Atp5c1*) (*Tappenden et al., 2011*), and cytochrome c oxidase enzyme (*Cox4i1, Cox7a*). ATP5a1 and ATP5c1 are important subunits of ATP synthase that regulate ATP synthesis. This suggests that the upregulation of these two genes signals the MPC1-deficient mice's constant demand for energy to survive. COX is the proposed rate-limiting enzyme; it contains several subunits and is involved in the electron transport complex, suggesting a regulatory role in modulating energy metabolism (*Huttemann et al., 2012*).

## MPC1$^{\pm}$ mice exhibit enhanced lipolysis and reduced lipogenesis

To further examine the effects of MPC1 deficiency on fat oxidation, we investigated the mRNA levels of several related key genes. Figure 4A shows that the relative mRNA abundances of key markers of lipolysis *hormone-sensitive lipase* (*Hsl*), *adipose triglyceride lipase* (*Atgl*), *cytochrome c oxidase subunit 4* (*Cox4*), and *perilipin2* were significantly

increased in the eWATs of the MPC1$^{\pm}$ mice. The release of metabolic energy, in the form of fatty acids, is controlled by a complex series of interrelated cascades that result in the activation of TG hydrolysis. The primary intracellular lipases are ATGL and HSL. Each TG molecule in the lipid droplets can be hydrolyzed to three FAs by ATGL and HSL. *Perilipin2* plays a positive role in adipocytes during lipolysis and modulates lipid absorption (*Takahashi et al., 2016*). *Cox4* is a gene involved in the electron transport complex, suggesting a regulatory role in modulating energy metabolism (*Huttemann et al., 2012*). Oxidation of fatty acids occurs in the mitochondria. The transport of long-chain fatty acyl-CoA into the mitochondria is accomplished via an acyl-carnitine intermediate, which itself is generated by the action of carnitine palmitoyltransferase 1 (CPT1) and carnitine palmitoyltransferase 2 (CPT2), enzymes that are present in the outer mitochondrial membrane. The *cpt2* gene was upregulated in the eWATs, whereas the *cpt1* gene was upregulated in the iBATs (Figs. 4A and 4B), thereby promoting the transport of fatty acids to the mitochondria for oxidation. *Cox4* was also upregulated in the iBATs (Fig. 4B).

Conversely, the transcriptional level of *fatty acid synthase* (*Fasn*) significantly decreased in both the iBATs and eWATs, and *acetyl-Coenzyme A carboxylase* (*Acc*) was significantly decreased in the iBATs. ACC is the most highly regulated enzyme in the fatty acid synthesis pathway. FASN catalyzes successive reactions to form fatty acids. Decreased mRNA levels of *acc* and *fasn* result in a reduction in lipogenesis. A previous study demonstrated that the disruption of MPC1 activity attenuates the accumulation of lipid droplets in MPC1$^{\pm}$ mice, resulting in low body weight and reduced eWAT (*Zou et al., 2018*).

We subsequently quantified protein enrichment in the MPC1$^{\pm}$ mice by western blotting. Figure 5 shows that the expression of ATGL, one of the key markers in lipolysis, significantly increased ($P < 0.05$) in the eWATs and iBATs of the MPC1$^{\pm}$ mice. The expression of this marker has also been associated with increased TG oxidation in mitochondria. As previously observed, the protein expression levels of one important synthase, ACC, were reduced in the eWATs and iBATs of the MPC1$^{\pm}$ mice, with that in the eWATs showing a statistically significant difference ($P < 0.05$). ACC is the rate-limiting step in fatty acid synthesis.

## DISCUSSION

Pyruvate lies at the intersection of glycolysis, gluconeogenesis, and the TCA cycle. Pyruvate is transported into the inner mitochondrial matrix by the MPC protein carrier. During normal cellular metabolism in the presence of the MPC complex, pyruvate is imported into the mitochondria where it is oxidized to acetyl-CoA which then enters the TCA cycle to produce reducing equivalents for oxidative phosphorylation. When MPC1 was knocked down, glucose metabolism in the TCA cycle was significantly, but not completely, decreased. A report showed that ATP-linked respiration is affected when all three pathways, namely, pyruvate transport, glutamine, and fatty acid oxidation, are inhibited (*Vacanti et al., 2014*). Alterations in glutamine levels have been detected by many researchers, whereas effective changes in fatty acid metabolism in MPC1$^{\pm}$ mice has received less attention but its relevant has lately been recognized (*Gray et al., 2015*; *McCommis et al., 2015*; *Vigueira et al., 2014*). *Vacanti et al. (2014)* observed that oxidative TCA flux was

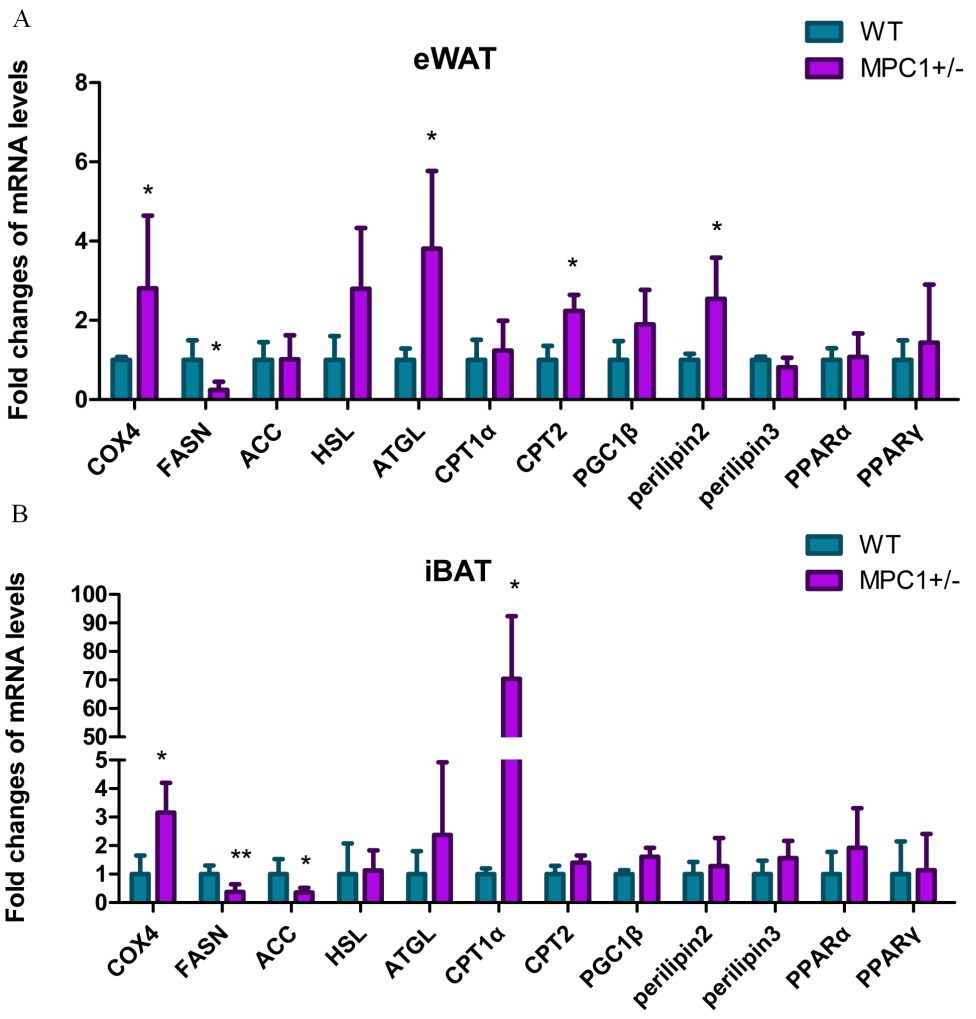

**Figure 4** **Energy metabolism-related genes influenced by MPC1 protein deficiency.** The mRNA expression of selected genes involved in energy metabolism pathway: fatty acid oxidation related genes such as *COX4, HSL, ATGL, CPT1α, CPT2, PGC1α, Perilipin2,* and *Perilipin3*; lipogenesis-related genes, including *FASN* and *ACC*; and energy metabolism-regulating genes such as *PPARα and PPARγ*. (A) Fold-changes of mRNA levels in the eWATs; (B) Fold-changes of mRNA levels in iBATs, values are expressed as the mean ± standard deviation, $n = 5$ *$P < 0.05$, **$P < 0.01$. All the mRNA levels were analyzed by qPCR.

achieved through enhanced glutaminolysis. The present study conducted mRNA-Seq, which indicated that the glutaminolysis pathway was upregulated, which agrees with the findings of our previous research that glutamine is oxidized in MPC1 deficiency. Upregulation of the *phosphogluconate dehydrogenase* (*Pgd*), *gamma-glutamyltransferase1* (*Ggt1*), *glutathione peroxidase 5* (*Gpx5*), *glutathione reductase* (*Gsr*), and *glutathione s-transferase* (*Gstm*) genes was also observed. These genes can elevate glutamine anaplerosis and oxidation.

Given the MPC position at the interface between mitochondrial pyruvate metabolism and glycolysis, it seems likely that alterations in MPC activity mightbe has relationship with tumour (*McCommis & Finck, 2015*). The results of this research indicate that the deficiency

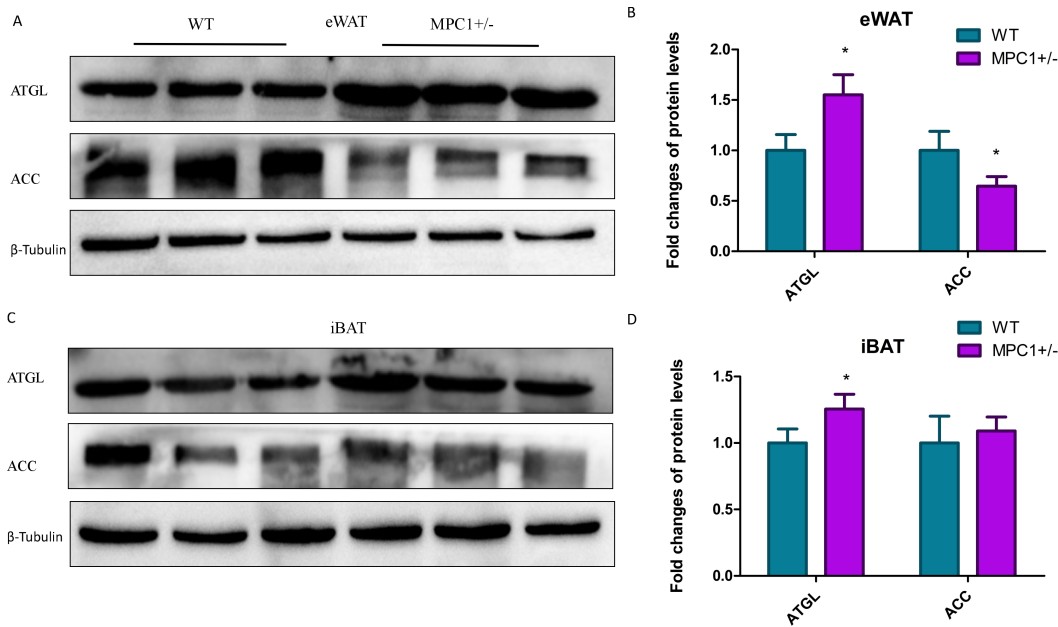

**Figure 5** **Western blot analysis.** (A) ACC and ATGL protein expression levels in eWATs; (B) The relative expression levels of ACC and ATGL protein in the eWATs; (C) ACC and ATGL protein expression levels in the iBATs; (D) The relative expression levels of ACC and ATGL protein in the iBATs; $n = 3$, the relative protein expression data were normalized to that of $\beta$-tubulin. $^*P < 0.05$, $^{**}P < 0.01$ for MPC1$^{\pm}$ mice *vs*. WT mice.

of MPC1 influences the expression of various genes that are involved in in the metabolism of cancer. Multiple mechanisms contribute to metabolic derangement in cancer, but the synthesis and metabolism of pyruvate play central roles (*Bayley & Devilee, 2012*). Low MPC1 expression is correlated with poor survival in almost all cancers (*Schell et al., 2014*). Similarly, chemical inhibition of tumor xenografts with CHC slightly enhances tumor growth (*Yang et al., 2014*). In an epidemiological investigation, Schell et al. demonstrated that the MPC1 is positively associated with cancer risks when the gene was underexpressed or deleted. In addition, MPC1 is downregulated in a variety of tumor cell lines and solid tumors, including those of the bladder, breast, brain and central nervous system, cervical cancer, colorectal cancer, esophageal cancer, gastric cancer, head and neck cancer, kidney cancer, liver cancer, lung cancer, ovarian cancer, pancreatic cancer, and prostate cancer (*Hong et al., 2007*; *Schell et al., 2014*). Although no cancer or tumor was observed in MPC1$^{\pm}$ mice, the mRNA expression levels of genes metabolized in hepatocellular carcinoma cancer, breast cancer, gastric cancer ,and prostate cancer were found to be abnormal.

Adipose tissues are the central metabolic organs involved in the regulation of whole-body energy and metabolic homeostasis, and the white adipose tissues function as key energy reservoirs (*Choe et al., 2016*). Considering the critical role of mpc1 in controlling energy metabolism (*McCommis & Finck, 2015*) and the relevant phenotype in white adipose tissue (*Zou et al., 2018*), we conducted various experiments using adipose tissues of heterozygous

(MPC1$^{\pm}$) mice. We did not perform much whole-body research except for the exploration of body weight and serum TG and FFA levels. There are certain limitations on the adipose tissue metabolic data in the absence of additional whole-body data. Adipose tissues act as organs that are involved in the regulation of glucose homeostasis and energy homeostasis via multiple metabolic signaling pathways targeting the liver, pancreas, skeletal muscle, and other organs (*Kim, Cho & Kim, 2014*). Though the release of adipokines, adipose tissue establishes a crosstalk with these organs (*Romacho et al., 2014*). On the other side, factors released from these organs can interact with adipose tissue itself, such as the myokines from the skeletal and the cardiokines from the heart (*Romacho et al., 2014*). The multidirecttional network of these organs may have interaction effect with the adaptive utilization of adipose tissues (*Ouwens et al., 2010*). Some fat depots such as the perivascular adipose tiuuse may additionally contribute to the complexity of interorgan crosstalk (*Ouwens et al., 2010*; *Romacho et al., 2014*). Increased fatty acid oxidation in skeletal muscle and reduced glucose production in liver will influence the secrete function of adipose tiuuses (*Dadson, Liu & Sweeney, 2011*; *Ouwens et al., 2010*; *Romacho et al., 2014*). In our next research, we plan to examine the metabolic status of the whole body. To uncover the function of mpc1 in energy metabolism, more studies involving cells and white adipose tissues using a knockout model are warranted.

In our previous study, we generated MPC1$^{\pm}$ mice and analyzed their phenotype when MPC1 protein was partially deficient, and the results indicated that the mice exhibited changes such as low body weight, decreased movement, and low body shell temperature, and a decrease in fat accumulation (*Zou et al., 2018*). Here, we detected molecular changes in adipose tissues (iBATs and eWATs) that are involved in energy metabolism, including lipogenesis and lipolysis. The MPC1$^{\pm}$ mice showed weak glycol metabolism, and carbohydrates were not able to generate sufficient amounts of ATP via mitochondrial respiration to sustain daily energy requirements. In response to these energy demands, stored fatty acids are mobilized to supplement energy. The low body weight and reduced adipose tissue accumulation that we observed in the MPC1$^{\pm}$ mice suggested that fatty acid oxidation was mobilized when mitochondrial pyruvate transport was limited. We provide a brief review of energy metabolism of fatty acids in an MPC1-deficient mouse model, focusing on the pathway of FA lipolysis (Fig. 6). Our intention was to provide a framework of the generation of new ideas on how to manipulate fatty acid metabolism in MPC1$^{\pm}$ mice. According to our results, MPC1$^{\pm}$ mice sustain the energy balance in mitochondrial metabolism by increasing the fatty acid $\beta$-oxidation and decreasing the fatty acid synthesis. Increased fatty acid metabolism and decreased lipogenesis result in low TG accumulation in all tissues, especially lipid droplets. In conclusion, the present study provides further evidence for the role of MPC1 in the energy metabolism. Screens to explore the metabolic mechanism of mpc1 in energy homeostasis may provide novel therapies for chronic diseases.

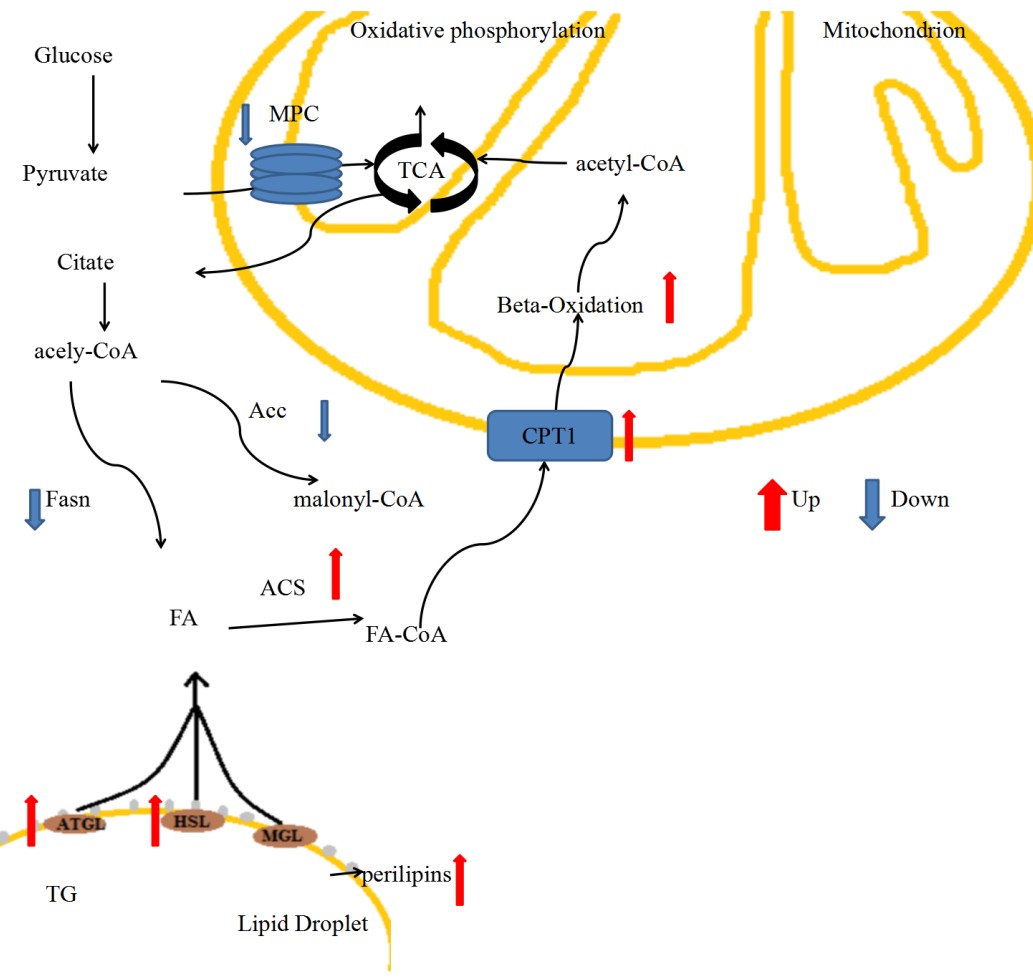

**Figure 6    Schematic changes through fatty acid metabolic pathways in MPC1± mice.**

### Funding

This work was supported by the Genetically Modified Organisms Breeding Major Projects of the People's Republic of China (2016ZX08011-005). There were no additional external funding received for this study. The funders had no role in study design, data collection and analysis, decision to publish, or preparation of the manuscript.

### Grant Disclosures

The following grant information was disclosed by the authors:
Genetically Modified Organisms Breeding Major Projects: 2016ZX08011-005.

### Competing Interests

The authors declare there are no competing interests.

# PeerJ

## Author Contributions

- Shiying Zou conceived and designed the experiments, performed the experiments, analyzed the data, contributed reagents/materials/analysis tools, prepared figures and/or tables, authored or reviewed drafts of the paper.
- Liye Zhu performed the experiments, contributed reagents/materials/analysis tools, authored or reviewed drafts of the paper.
- Kunlun Huang approved the final draft.
- Haoshu Luo performed the experiments.
- Wentao Xu conceived and designed the experiments, approved the final draft.
- Xiaoyun He conceived and designed the experiments, analyzed the data, prepared figures and/or tables, authored or reviewed drafts of the paper, approved the final draft.

## Animal Ethics

The following information was supplied relating to ethical approvals (i.e., approving body and any reference numbers):

This animal study was approved by the Animal Experimental Welfare & Ethical Inspection Committee (2016005), the Supervision & Testing Center for GMO Food Safety, Ministry of Agriculture (Beijing, China).

## Data Availability

The raw data are included in the Supplemental Files.

## Supplemental Information

Supplemental information for this article can be found online at http://dx.doi.org/10.7717/peerj.5799#supplemental-information.

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
