# Peer review of "Adipose tissues of MPC1± mice display altered lipid metabolism-related enzyme expression levels"

_PeerJ, doi:10.7717/peerj.5799_

## Round 0.1 · original submission · Major Revisions

All of our reviewers detected substantial problems which must be addressed before your paper can be accepted. I urge you to take special care in the following points:

-ensure that blots shown have not been flipped vertically ( as seems to have happened with ATGL from eWAT in MPC1+/-. As requested by reviewer #1, do provide proper legends to every lane in the uncropped blots deposited as Supporting Material.

-include the mitochondrial markers and immunoblots requested by reviewer #2

- provide the phenotypic data requested by reviewer #3, as well as photos of tissues from different animals and statistical data from a larger number of animals. To provide a better understanding of you manuscript's focus, please discuss the main similarities/differences between this study and Zou et al. 2018
* * *
When you prepare your rebuttal, provide the full text of all of the reviewers' comments to the initial version of this submission, interspersed with your detailed replies to each point (preferably in a different font, for ease of reading).

PeerJ requests that re-submissions be accompanied by a copy of the manuscript file with highlighted changes. Do not highlight those changes manually: use your word-processor built-in "track changes feature" instead, to compare the initial submission to your modified manuscript.

·

Basic reporting

The manuscript by Zou and colleagues reports on the phenotype of mouse heterozygote for the MPC1 subunit of the mitochondrial pyruvate carrier and analyzes the transcriptome of epididymal fat (eWAT) and interscapular brown adipose tissue (iBAT) by RNA-seq. Following observation of reduced eWAT size, the transcriptome analysis revealed that genes coding for fatty acid metabolism were deregulated in MPC1 +/- adipose tissue. Changes in specific genes were confirmed by qPCR and western blotting to show that proteins involved in lipogenesis were down regulated and proteins essential for lipolysis and lipid oxidation were up regulated in MPC1+/- eWAT and iBAT. Overall, the manuscript is clear despite a few specific points that need to be addressed, as described bellow.
- Abstract: line 21, Use of "Despite" is confusing as the phenotype of smaller adipocyte is consistent with decreased eWAT and body weight.
- Line 36: The original paper showing that MPC1 is essential for embryonic development (Vanderperre et al. 2016) should be cited here rather than later in the introduction.
- Line 40: typo. "pregnant does" should be "pregnant dams"
- Line 59 eWAT is the largest visceral "WAT" not the largest visceral eWAT as stated.
- Line 178: The sentence about eWAT activation is confusing and does not relate to anything analyzed in this study. Consider removing the sentence.

Experimental design

The experimental design is appropriate. The validation by qPCR and western blotting of changes identified by RNA-seq reinforces the validity of the findings.
However, there are several issues with figures that need be fixed and in some cases, better explained, as listed bellow.

- Figure 4, the legends should be more detailed. For example by stating that mRNA levels were analyzed by qPCR. In addition, panels A and B are not identified in the figure.
- Figure 5, panels are also not identified in the figure and the names of protein are misaligned with blots.
- Figure 5, pannel A: the actin blot is the same in WT and MPC1+/- samples. Based on the uncropped images provided as supplement, it appears that this is an honest mistake by the authors as they do have an actin blot that contains 6 lanes as the other blots.
- Figure 5: the uncropped images show 6 lanes for each western blot, but the author chose to show only 4 samples. While there may be a good reason for this, it raises questions about why the author pick only some samples to show. More disturbing is the fact that it appears that they do not show the same samples for the different antibodies. The ACC blot appears to be samples number 1,2,5 and 6 while the ATGL blot shown samples 2,3,4 and 5. Can the author explain what these 6 samples are? In addition, an n of 3 for mice samples should be the minimum to analyze levels and provide statistics. I suggest the authors generate an n=3 for the western analysis and show all 6 samples in a contiguous image (rather than separated boxes for WT and MPC1+/- as currently shown) and repeat the level analysis on 3 samples for each genotype.
- Finally, the method used for quantifying the western blot signal should be described.

Validity of the findings

The mouse model used in this study was generated using CRISPR/Cas9-mediated gene knockout. Based on the reference provided, it appears that the gRNA sequence used to delete the MPC1 allele has several predicted off-target effect with 0 or 1 mismatch, which makes it more likely that these sites could also be mutated in these mice. The authors should provide targeted sequencing data for the most likely predicted off-target sites to show that these sites were not mutated in the founder animal.

Line 193, their data do not suggest that TG have been metabolized by peripheral tissue as opposed to decreased TG synthesis. This sentence need be better identified as speculation.

Additional comments

no additional comments

Reviewer 2 ·

Basic reporting

See below

Experimental design

See below

Validity of the findings

See below

Additional comments

In the manuscript "MPC1 deficiency enhances the metabolism of triacylglycerol in MPC1+/- mice", Zou et al show that heterozygous deletion of the gene for mitochondrial pyruvate carrier protein 1 in mice increases the expression of genes involved in the catabolism of triglycerides in epididymal white and intrascapular brown adipose tissue. The morphology and large-scale gene expression pattern were analyzed in these two tissues of MPC1+/- mice and compared to wild-type mice. Although the manuscript contains some interesting results there are vital control experiments which are lacking (see comments below). Therefore, the manuscript in its present state is not recommended for publication in PeerJ.

Major comments
The title of the manuscript should be changed to reflect directly the results obtained, such as for example: "Adipose tissues of MPC1+/- mice display altered expression levels of enzymes involved in lipid metabolism"

Fig. 2. A mitochondrial marker should be used to detect differences in the number or volume of mitochondria in eWAT of MPC1+/- mice with respect to WT.

Fig. 5. As the whole manuscript relies on the assumed lower expression level of MPC1 protein in eWAT and iBAT of MPC1+/- mice with respect to WT, this should be demonstrated by immunoblotting using material from these tissues.

A supplementary Table listing all the genes of Fig. 3 should be included for data transparency and to show in which background the genes of interest were selected.

Minor comments
Line 17 says "the liver", but no analysis on liver is described in the manuscript.

Line 19. The meaning of "an MPC1 protein part deficiency" should be clarified.

Lines 20 and 170. The meaning of "heterozygous mice exhibited low energy metabolism" should be clarified.

Line 21, says "Despite". Maybe "In concomitance with" is more appropriate.

Fig. 4. Probably the abbreviations "AGTL" in the graphs should be "ATGL".

Fig. 5. The use of "relative expression level" in the graphs should be explained (relative to what?).

The paragraph on the connection between MPC1 and cancer (lines 313-326) should be rewritten and made more clear.

All the figure legends should be revised to provide the complete information required for the reader to understand the figures.

The manuscript needs to be improved by using English language help for overall clarity and removal of mistakes.

Reviewer 3 ·

Basic reporting

See general comments

Experimental design

See general comments

Validity of the findings

See general comments

Additional comments

This is a straightforward and simple study that examines the effect of whole-body MPC1 knockout on adipose tissue metabolism. The authors show that MPC1 +/- mice are smaller and have reduced adipose tissue mass, and changes in the expression of adipose tissue metabolism genes. The paper is adequately written, clearly laid out and the results are presented clearly. However, the data are extremely preliminary and very limited in their scope, most likely because the authors have very recently published a related manuscript containing some of the basic phenotypic data that should be included here (Zou et al., 2018). I have a few additional specific comments:

1) Clearly the major weakness of the paper is that the authors did not analyze the phenotype of adipose tissue-specific MPC1 knockout mice. This is important as it is known that whole-body MPC1 knockout mice are embryonically lethal. Moreover, the MPC proteins have important roles in liver metabolism and are critical for brain (and probably adipocyte) development. Together this makes the interpretation of the MPC1 +/- mouse data very difficult, if not impossible. If the authors are intent on pursuing studies in this whole-body model, the authors need to address the following points regarding the very basic analyses of the MPC1 +/- mice phenotype:

i) Where is the quantification of MPC1 and MPC2 protein levels in the MPC1 +/- mice? This is essential to perform. Do the authors have these data? Knockout of MPC1 usually leads to the degradation of MPC2 protein, and vice versa. I only see mRNA data in figure 4, but this is not very convincing.
ii) Were there changes in liver, muscle, heart and pancreas mass? Did the authors perform whole-body qMRI analysis?
iii) What about whole body energy balance, including food intake and energy expenditure?
iv) Did the mice have changes in glucose metabolism? The authors could examine fasting plasma glucose levels (at a minimum) very easily. Similarly it would be important to quantify circulating triglycerides and/or FFA levels. This can also be done easily and cheaply.
v) I note that some, but not all, of these data are presented in a related manuscript by the same group (Zou et al., 2018). It would have been more impactful to combine these data into one larger, more comprehensive analysis. A brief look at the data in this manuscript shows the expected abnormalities in whole body RER, hepatic glucose metabolism etc. This reinforces my concerns about focusing on a specific tissue in the context of a whole-body knockout.

2) In figure 1, only 6 mice were examined in each group. This is quite a small number of mice for these studies. Photos of adipose tissue should be shown from more than one mice in my opinion. In the previous paper (Zou et al., 2018), 6 mice were also used for all studies. Are these the same six mice shown here?
3) In figure 2, the authors should improve their presentation of the adipocyte size distribution. The axes should be swapped and the number of adipocyte diameter bins should be expanded so that the overall distribution of adipocyte size can be better appreciated.
4) In figure 5, only two mice were used for quantification and statistical analysis. This is inappropriate and more samples should be analyzed to enhance the rigor of the studies. The provided “uncropped” Western blot images look like cut film to me. Do the authors have whole-films with molecular weight markers included?
5) The discussion does not address the weaknesses of the study adequately. Specifically the issues relating to point 1 above are not introduced. This would be important to do in a revised article.

---

## Round 0.2 · Major Revisions

Reviewer #3 still has serious misgivings regarding the presentation o fadipose tissue metabolic data in the absence of additional whole-body data. Please discuss thoroughly the possible limitations entailed by this.

·

Basic reporting

see below

Experimental design

see below

Validity of the findings

see below

Additional comments

The authors have addressed all my comments. With the exception of a few text edit required as described below, the manuscript in sound and should be accepted for publications.
Minor edits:
Line 21-22: Should use "consistent with" rather than "In consistent"
line 221: should say 1625 genes "with" a significant difference to WT

Reviewer 2 ·

Basic reporting

.

Experimental design

.

Validity of the findings

.

Additional comments

The revised version of the manuscript now entitled "Adipose tissues of MPC1+/- mice display altered expression levels of enzymes involved in lipid metabolism" (26773-v1) by Zou et al. has been improved and my previous comments have been dealt with properly. However, it still needs English language editing to improve clarity and remove errors. Furthermore, new errors have been introduced in the revised version.

Reviewer 3 ·

Basic reporting

See general comments

Experimental design

See general comments

Validity of the findings

See general comments

Additional comments

My concerns about the interpretation of the adipose tissue findings in relation to the whole-body knockout phenotype have not been addressed in this revised article. Many of the findings in the adipose tissue could be due to primary effects elsewhere (liver, muscle, pancreas), and potentially on the development of the adipose tissue (and other tissues). I think presenting the adipose tissue metabolic data in the absence of additional whole-body data is misleading.

---

## Round 0.3 · Minor Revisions

I am afraid that the text you added to the conclusions (detailing possible limitations of the study) is insufficient both in content and in language quality (e.g. ref. missing in"According to the results of our early study, we found low energy consumption [...]"; the simple enumeration of possible limitations without any references or further discussion; sentences (or sentence fragments) without indicative mood verbs; etc.). Please revise its contents and then ensure its linguistic adequacy by running it through a professionally-proficient English speaker.

---

## Round 0.4 · Minor Revisions

I am afraid I must insist on language revision by (preferably) a professional service, as the abundance of grammatical mistakes, flow interruptions, and bad punctuation seriously impede the enjoyment of your manuscript. I also suggest that you rearrange the order of insertion of the recently-added paragraphs in th conclusion section. Please be extremely careful and thorough, to prevent the need for additional rounds of revision. Content-wise, I think that the limitations are not yet adequately addressed: for example in line 362 you state "The metabolic data in these organs may have interaction effect with the adaptive utilization of adipose tissue. " but do not provide any supporting references or description of the known factors underlying those interactions.

---

## Round 0.5 · accepted · Accept

I think the changes you introduced are satisfactory.

#